# Modelling Development, Territorial and Legislative Factors Impacting the Changes in Use of Agricultural Land in Slovakia

**Lucia Palšová [1,\*], Katarína Melichová [2]**  **and Ina Melišková [1]**

[1] Department of Law, Faculty of European Studies and Regional Development, Slovak University of Agriculture in Nitra, Tr. A. Hlinku 2, 949 76 Nitra, Slovakia

[2] Department of Public Administration, Faculty of European Studies and Regional Development, Slovak University of Agriculture in Nitra, Tr. A. Hlinku 2, 949 76 Nitra, Slovakia

\* Correspondence: lucia.palsova@uniag.sk; Tel.: +421-37-641-5079

**Abstract:** The conflict of interests in agricultural land use based on the diversity of needs of private and public interest is the main problem of the current protection of agricultural land in Slovakia. Therefore, the aim of the paper is to identify factors affecting the withdrawal of agricultural land, i.e., conversion of the agricultural land to non-agricultural purposes, and to initiate a professional discussion on the concept of protection and use of the agricultural land in Slovakia. Through panel regression models, the developmental, territorial, and legislative factors affecting land withdrawal for the purpose of housing, industry, transport, mining, and other purposes were analyzed. Research has shown that developmental factors, compared to legislative ones, affect the total volume of agricultural land withdrawn in bigger scope. From the perspective of the conflict of interests between the individuals and state regarding land protection, the private interest prevails over the public one. As a consequence, agricultural land is withdrawn in suburbanized and attractive areas, where the land of the highest quality is mostly located. In accordance with the precautionary principle, the state should adopt a long-term conceptual document defining the areas of agricultural land use taking into account the impact of the developmental factors on the land protection.

**Keywords:** withdrawal of agricultural land; contributions; developmental factors; territorial factors; legislative factors

## 1. Introduction

The competition for land resources creates serious risks of geopolitical imbalances in the European Union (hereinafter as "EU") and worldwide [1]. For the following reasons, it is currently emphasized to ensure effective tools of land policy. Its complexity lies in the fact that the management of land nearly always faces a trade-off between various social, economic, and environmental needs [2]. Land use for non-agricultural purposes has resulted in impermeable coverage of the land as an irreversible process leading to the loss of soil resources [3–8].

Research on land use is now being redefined. Until now, the emphasis had been on the impermeable coverage of land in general [9–11], currently more important is the examination of the factors causing the land withdrawal. Indeed, the withdrawal of agricultural land for non-agricultural purposes will often irreversibly depreciate the most productive soil in the area, and therefore there is an emphasis put on the support for the rational decision-making process of the administrative authorities [12–14].

The process of agricultural land withdrawal is predominant in suburban areas [7,8,15]. In some countries, due to the rapid urbanization process and the increasing value of land, developers started to focus on the construction of housing units on low quality land [16].

The issue of urbanization and the withdrawal of agricultural land is a worldwide phenomenon, and while it draws attention to the economic growth, the majority of the agricultural land is withdrawn for non-agricultural purposes (e.g., up to 70% in Pakistan), resulting in a lack of food and nutrition for the growing population and an increase of socio-economic and infrastructure issues [17]. China is another example, where they are experiencing severe depletion of agricultural land in several urban agglomerations. This situation underlines the urgent need for the government to develop an effective policy for the protection of agricultural land [18–21]. An elaboration of studies on the use and withdrawal of agricultural land can be seen as an important tool for policy makers and future generations and for maintaining ecological well-being of inhabitants [22]. A study conducted in selected EU cities has demonstrated that an integrated approach, along with other sustainability objectives instead of isolated programs, reduces the withdrawal of land and subsequent impermeable coverage [5].

The specificities of the land use conflicts in the EU are also linked to the implementation of the Common Agricultural Policy (hereinafter as "CAP"), which affects the regional land market by disrupting agricultural land prices. Research in Poland has shown that in the case of the most urbanized regions, the contribution of the CAP to the agricultural land prices was relatively low, leading to a strong incentive for farmers to sell their land for non-agricultural purposes and thus to land use conflicts [23]. Contrary, in Italy, the risks associated with changes in the use of agricultural land are the most significant. Its consumption is mainly related to the urbanization pressure, calling for the highly efficient decision-making tools, such as municipal town planning schemes [24,25]. All EU countries are currently addressing the withdrawal of agricultural land related to the urban expansion, with the objective to help authorities and policy makers understand the effects of urban expansion and related land use change on agriculture [26,27]. Research in this area has shown that the CAP subsidies have an impact on reducing urbanization and reducing agricultural land use [28].

Despite the importance of joint action to protect the area and quality of the agricultural land in the EU, responsibility for land policy and legislation and its implementation remain a national competence of the EU Member States [29].

Slovakia is the EU country with the smallest total area of agricultural land (48.58% of agricultural land, of which 50% is arable land [30]). The majority of the agricultural land belongs to the quality categories 5–7 (out of the total 9 quality categories ordered from the most to the least productive [31]).

The conflict between satisfying the private and public needs is addressed by the state through the state legislative instruments. The protection of agricultural land is defined as the basic constitutional obligation of the state [32], which was subsequently amended in the sectoral Act No 220/2004 Coll. on the protection and use of agricultural land and amending Act No 245/2003 Coll. on integrated pollution prevention and control and in amendments to certain acts (lex generalis).

From a qualitative point of view, the Act transfers the state's responsibility for protecting the land to the owner, user or land manager. Although the law expressis verbis determines the obligations of the parties concerned, enforcement for the infringement is weak [33]. The reason is the legal and factual problem related to the complexity of the whole issue (causality and its proving, reversibility of damages, forms of remedy) on the one hand and the small societal and political interest in increasing the enforceability of legal obligations on the other hand [34].

From a quantitative point of view, the Act regulates the withdrawal of land for non-agricultural purposes according to the framework areas of the occupation, but does not reflect the purpose of the land. The Act is limited to economic instruments, such as contributions for the withdrawal of agricultural land for non-agricultural purposes and fines for breaching of land area protection obligations. The contributions are based on the principle that the person in whose interest the land is withdrawn is obliged to pay a specified amount [5]. Therefore, they have a motivation function, as it should lead the applicant to withdraw the land to limit the required occupation, to choose the withdrawal of land of less quality, or choose an alternative solution [34,35].

Also for this reason, contributions have become an effective tool of the state to pursue state land policy objectives. The development of legislative changes implies that the state has abolished, respectively modified, the contributions according to the expected investment development. With the accession of Slovakia to the EU in 2004, the contributions were abolished in order to open up the internal market [36]. As the legislator's expectation of a positive impact on landowners or users has not been met, in 2009 contributions were reintroduced for the agricultural land of the highest quality [37]. In 2013, contributions were extended to withdrawal of agricultural land of the highest quality in each cadastral area [38]. The introduced change has brought a positive effect on the protection of the land area and balanced development within the regions [39]. Constitutional change in 2017, brought a special protection of agricultural land [32].

The application of the Act is left to the decision-making competence of the employees in state administration on the district level, who lack methodological tools and clear guidelines on how to solve conflict in land use [40]. Izakovičová et al. [2] provide such a methodical approach to the integrative assessment of the land use conflicts specifically under conditions currently present in Slovakia.

In Slovakia, the territorial concept of the use of agricultural land and the planning of strategic activities has been absent for a long time. This assumes an analysis of factors affecting the withdrawal of agricultural land and the subsequent establishment of long-term priorities, which takes into account indicators of soil quality, climate change and development [41,42].

As the impacts of the state land policy on the sustainable use of agricultural land have never been comprehensively examined in Slovakia, the aim of the article is to identify development, territorial, and legislative factors affecting the loss of volume of agricultural land in Slovakia due to its withdrawal.

## 2. Materials and Methods

To achieve the aim, first, we conducted an analysis of the spatial distribution of the withdrawal of agricultural land in Slovak districts for the period of 2007–2016, due to data availability. To better comprehend trends in the agricultural land withdrawal, a Delphi method was employed. The panel of experts consisted of 28 respondents from the relevant scientific fields as well as from practice (Ministry of Agriculture and Rural Development of the Slovak Republic, Soil Science and Conservation Research Institute, Research Institute of Agriculture and Food Economics, Central Control and Testing Institute in Agriculture, Slovak University of Agriculture in Nitra, chairman of an agricultural cooperative and representatives of the Land and Forestry Departments of district offices located in the seats of 8 NUTS III regions of the Slovak Republic).

Conducting the interviews with listed experts allowed us to confront the empirical findings with the information provided by the experts. Consequently, this confrontation, together with the theoretical assumptions discussed in detail in the first part of the paper, helped us to narrow down the factors affecting the volume of agricultural land withdrawal, as well as to formulate the final statistical models used to quantify their effects statistically. For this purpose, in the final part of the paper, a panel data set containing the information on the amount of land withdrawn from the agricultural land fund (for housing development purpose, industrial development purpose, mining, transportation and other purposes) for the period of 2007–2016 and spanning 41 districts of Slovakia was used, meaning 410 observations in total. The withdrawal of agricultural land is an administrative process that results in the conversion of agricultural land into one of the alternative uses listed above.

The decision to extend the analysis to panel data (as opposed to a simple time series) was reached mainly due to the findings generated by the interviews with experts, especially the need to take into account regional and local specificities. Indeed, interviews with representatives of the district authorities revealed that the factors and circumstances that caused the intensity of the withdrawal of agricultural land in individual districts varied spatially. However, this decision eliminated a group of potentially important factors, for which it is not possible to obtain relevant data on the district level. In addition, some important factors are invariable or do not vary sufficiently, resulting in an arbitrary decision to use a panel regression model with random errors. The dependent variable of the individual

models is the volume of agricultural land withdrawn, calculated as a proportion to the population of the respective district in a given year. We examine the effect of relevant factors not only on the total volume withdrawn but also on the volume withdrawn for individual purposes, since it is justified to assume that the circumstances affecting the intensity of land withdrawal will be different. Data on the withdrawal of agricultural land were obtained from the Electronic Land Service Yearbook and from the Ministry of Agriculture and Rural Development of the Slovak Republic.

Individual potentially relevant developmental, territorial, and legislative factors enter the models as explanatory variables and have been chosen based on the study of relevant literature in the first part of the article and based on the results of interviews carried out with the panel of experts.

All input data were standardized by calculation of *z*-scores using the formula:

$$I^s_{ij} = \frac{I_{ij} - \overline{I_j}}{\sigma}$$

where $I^s_{ij}$ is the new normalized value of indicator $I$ in district $i$ in year $j$, $\overline{I_j}$ is the average value of indicator $I$ in the year $j$ and $\sigma$ is the standard deviation.

In terms of the developmental factors, we examine the impact of foreign direct investment flows—FDI_POP (measured as foreign direct investments in euros per capita; data were obtained from the National Bank of Slovakia), and domestic business growth—DBUS_POP (measured as the number of newly established enterprises per capita; data were obtained from the Register of Organizations). Additionally, income level of the population—AMNW (measured as the average monthly nominal wage), the number of immigrants—IMMIG (measured as the absolute number of immigrants), the net migration—NETMIG and NETMIG_POP (measured as a migration balance in absolute terms and relative to the population in the base year) were included into the models and obtained from the Statistical Office of the Slovak Republic. In the case of migration, all three indicators need to be taken into account, as each reflects a different aspect of the impact of population movements.

In the case of territorial factors, the most important characteristic is usually the settlement structure, not only because it affects the demand for the use of space, but also because it is closely related to the developmental factors. In our analyses, we examine two: The average size of settlements in the district—AVER_SET (measured as the arithmetic average of the population of all municipalities and cities in a given district) and the size of the central city—CENT_CITY (measured as the population of the largest city in the district) for which the data were obtained from the Statistical Office of the Slovak Republic. In addition, we also consider the territorial characteristics of the land fund itself, namely the fragmentation of land. We also analyze two factors: Fragmentation of arable land—FRAG_ARABLE (measured as the average size of land blocks identified as arable land) and fragmentation of agricultural land—FRAG_AGRI (measured as the average size of land blocks identified as agricultural land). A shapefile of Land Parcel Identification System (LPIS) blocks was used to generate the variables, and it was obtained from the Ministry of Agriculture and Rural Development of the Slovak Republic. It should be noted that the land fragmentation data is only available for the present, so these variables have only a spatial dimension.

In the case of legal factors, while constructing the model, we rely mainly on the information provided by the panel of experts. Dummy and categorical variables quantifying potentially relevant legislative milestones were used. Dummy variable—EXEMP_VINEYARD was designed to introduce contributions on the withdrawal of vineyards as well as the introduction of exemptions for family homes, where 0 means the absence of the regulations in question and 1 means that the regulations are introduced/in force. These two legislative changes were introduced in the same year and were in effect until the end of the analyzed timeframe, so we cannot separate their effects. A categorical variable CON was used to measure the impact of contributions imposed for the withdrawal of agricultural land because in addition to being introduced in the given period, the contributions were also significantly altered (by adding land quality categories 5 to 9 later on). The reference value of the variable acquires a value of 0 if contributions were not levied for the withdrawal of land, 1 for the years when they were

paid only for quality categories 1 through 4 and 2, for the years when contributions were paid for withdrawal of all quality categories. In this case, it should be noted that the variables reflecting the legislative milestones have only a temporal dimension as they apply equally to all spatial units. For every withdrawal purpose we also examine all the factors together, with the resulting panel regression model having the following form:

$$
\begin{aligned}
zWTD_{it}^{p} = \quad & \alpha + \beta_1 zFDI\_POP_{it} + \beta_2 zDBUS\_POP_{it} + \beta_3 AMNW_{it} + \beta_4 zIMMIG_{it} \\
& + \beta_5 zNETMIG_{it} + \beta_6 zNETMIG\_POP_{it} + \beta_7 zFRAG\_ARABLE_i \\
& + \beta_8 FRAG\_AGRI_i + \beta_9 zAVER\_SET_{it} + \beta_{10} zCENT\_CITY_{it} + \beta_{11} CON(1\_4)_t \\
& + \beta_{12} CON(1\_9)_t + \beta_{13} EXEMP\_VINEYARD_t + u_t + \varepsilon_{it}
\end{aligned}
$$

where $zWTD_{it}^{p}$ is the value of the dependent variable in district $i$ in year $j$ (normalized value of volume of withdrawn agricultural land for purpose $p$), $\alpha i$ denotes the value of intercept, $\mu_i$ introduces a time-invariant district-specific unobserved component, while $\varepsilon_{it}$ is idiosyncratic error term.

The proposed approach is original mainly in its scope, both geographic and in terms of comprehensiveness of the impacts tested. In Slovakia specifically, analyzing the land use conflicts and subsequent changes relies on conducting case studies [2] or on using a specific set of indicators, like legal aspects [39]. We believe that analyzing both development and legislative factors, while taking territorial conditions into account, will help to determine how the individual entities and public policies interact in land use conflicts, thus providing insights into what needs to be done to increase the effectiveness of the legal framework aimed at achieving sustainable use of land resources.

## 3. Results and Discussion

In Slovakia, due to historical and territorial circumstances (the emergence of of separate state, socialism), emphasis was placed on the farming on agricultural land to the detriment of ownership relations with the aim of increasing production. As a result, despite the social and political changes in 1989, more than 90% of the agricultural land is currently leased, causing the loss of owners' relationship to the land and loss of their motivation to implement sustainable agriculture practices [41]. Therefore, when the demand for investment activities and the convergence to the EU-15 standard of living occurred, after the accession of Slovakia to the EU in 2004, landowners were not interested in maintaining the land in the agricultural land fund. In addition, through legislation, the state had created a leeway for land use for non-agricultural purposes for both private and public needs [43].

The trend of withdrawing of the agricultural land for non-agricultural purpose was fluctuating in the given period (Figure 1). The most pronounced conflict in alternative use of the agricultural land has been proven between both agriculture and residential use and agriculture and industrial development. The most notable extreme in amount of withdrawn agricultural land happened in 2008, which is the last year when the contributions for withdrawal were not levied on those that withdrew agricultural land of top four quality categories. An increase was recorded in withdrawal for all listed purposes, the most significant, however, occurred due to the conversion to residential and industrial use, as well as the accompanying infrastructure development.

Almost 5 thousand hectares of agricultural land were withdrawn in one year alone, clearly indicating that both private and public entities reacted to the upcoming legislative amendment introducing the contributions in the following year. In the following years, the trend of land withdrawal steadily declined, mostly not exceeding the level of one thousand hectares per year. Over the course of analyzed time period, two more rises occurred. The first one was in 2012, in the year that also coincided with relatively major legislative change, namely the extension of the contribution obligation for withdrawal of agricultural land for all quality categories that came into force the following year. This, however, seemed to affect mostly the withdrawal for transport infrastructure development. This is usually planned years ahead, so the causative relationship between the mentioned contribution extension and a prior increase in land withdrawal might not be straightforward. The second increase

occurred in 2016, which was caused by a marked rise in the conversion of agricultural land to industrial sites and it coincided with the latest major foreign investment in Slovakia – Jaguar Land Rover located in the Nitra district.

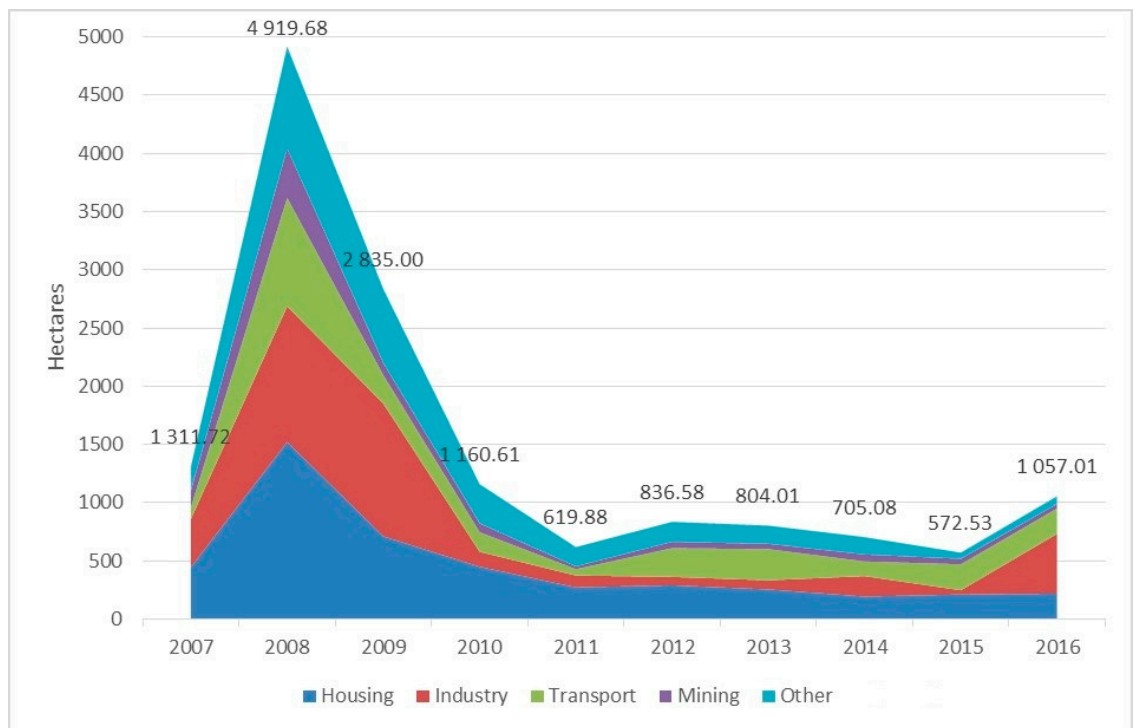

**Figure 1.** Volume of withdrawn agricultural land according to the purpose of withdrawal in the Slovak Republic in 2007–2016 (in ha). Source: Own processing based on the Electronic Land Service Yearbook and data from the Ministry of Agriculture and Rural Development of the Slovak Republic.

Overall, the lowest amount of the agricultural land was lost for mining purposes, which seems to be phasing out almost completely in recent years.

Based on the total amount of the withdrawn agricultural land for the analyzed ten-year period (Figure 2) we can conclude that the most intensive conflict in land use arises in the western and north-western part of the country, supporting the assumptions presented in Reference [42]. The greatest losses of agricultural land occurred in the districts with a heavy presence of foreign investments (Trnava, Nitra, and Galanta) located close to the capital city Bratislava. Over the ten-year period, the withdrawal in these districts was caused mainly by the expansion of industrial areas, most notably the automobile sector and expansion of the related economic activities, which was confirmed by the experts. Although the amount of the withdrawn agricultural land in the Bratislava district is relatively low, all neighboring districts experienced severe pressure in alternative use of agricultural land, with a most pronounced increase in demand for the residential space. Although the capital city of Bratislava, and the most developed region in Slovakia, exerts positive economic spatial spillovers on the surrounding regions, it also creates intense conflicts in the use of land, resulting in permanent losses of the agricultural land of the highest quality. We can observe a similar influence of Košice—the second largest city in Slovakia, located in the eastern part of the country, although the land takes intensity in its neighboring districts is of a lesser magnitude. Overall, in the eastern districts, aside from the development of industrial and commercial sites, residential use seems to be the biggest rival to the agricultural sector in use of land.

In the tables below (Tables 1–6), for each withdrawal purpose under analysis, we introduce several models representing different combinations of factors defined in the Materials and Methods section of the article. Specifically, Model I represents the estimates of the impacts of developmental factors

on the withdrawal of agricultural land, model II represents the estimates of the impacts of territorial characteristics, model III the impact of significant legislative changes, and model IV the combination of all the mentioned factors. Statistically significant regression coefficients are indicated by asterisks in the table, while standard errors are given in brackets below the regression coefficients. Due to the standardization of input data, the coefficients are dimensionless and can be compared.

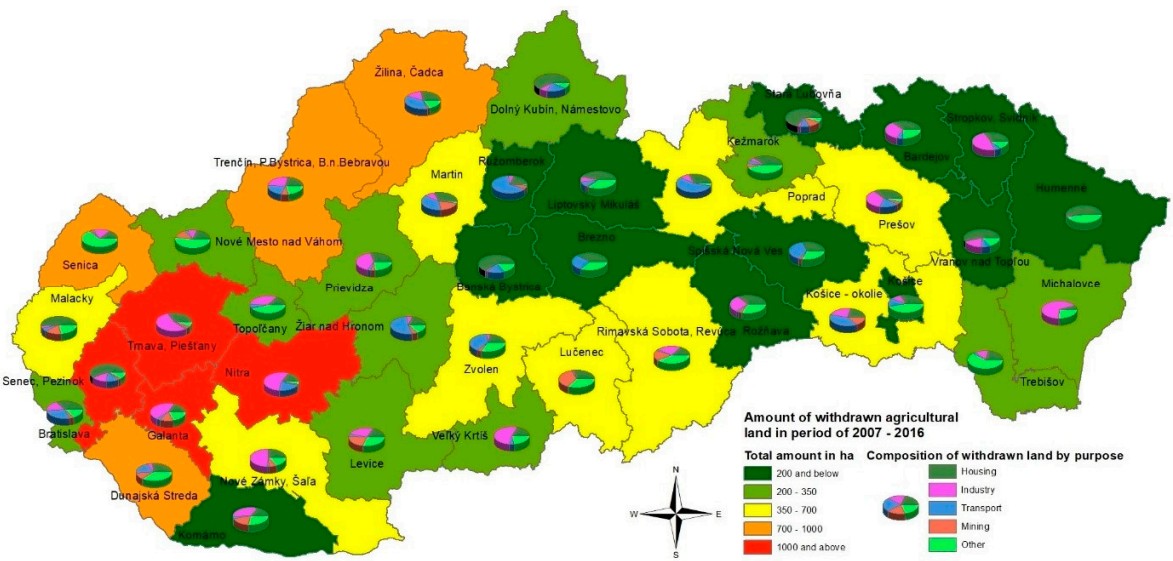

**Figure 2.** Spatial aspects of the agricultural land withdrawal in Slovakia in 2007–2016. Source: Own processing based on the Electronic Land Service Yearbook and data from the Ministry of Agriculture and Rural Development of the Slovak Republic.

From Table 1 it is clear that all factor groups have a statistically significant effect on the total volume of the agricultural land withdrawal (all models are statistically significant, as evidenced by Wald's $\chi^2$). By comparing the $R^2$ coefficients of determination, it can be concluded that the developmental factors have the greatest impact on the total volume of the agricultural land withdrawn. Among these, foreign direct investment, income level, as well as mechanical population movement (i.e., migration) are statistically significant. This confirms not only the theoretical assumptions but also the statements of several interviewed experts and previously conducted studies [44–46].

In the case of absolute value of the migration balance, we have identified contradictory effects. The increase of the absolute value of the migration balance leads to the reduction in the volume of the land withdrawal. However, when we recalculate the migration balance as a share of the total population in the base year, we find the effect has opposite direction. With the increase in the net migration per capita, the volume withdrawn from the agricultural land fund increases significantly.

Although suburbanization and de-urbanization processes are already taking place in Slovakia, the urbanized regions tend to have the highest migration balance. In many cases, these are the regions where the starting point of the agricultural land area has already been very low. It can, therefore, be assumed that the increasing demands for the alternative use of space caused by immigration are being addressed to a greater extent by adapting areas, other than agricultural land, to new functions for which demand has increased (e.g., former industrial districts, increasing the number of floors in residential areas, etc.).

However, when assessing the net migration balance relative to the baseline population number, the highest values of this indicator are recorded by the regions that previously had a lower population density but became more attractive to migrants at that time. Often these are the regions that are the target of suburbanization processes. Thus, we have also statistically confirmed that it is precisely in these regions that the most significant changes in the use and functions of the land are taking place (as existing capacities are not sufficient given the growing demand). Charles and Guna [47] affirm that

migration has a negative impact on the city's limited land and it is necessary to propose solutions for its sustainable use strategy.

**Table 1.** Results of panel regression models where the dependent variable is the total volume of agricultural land withdrawn.

| Factor: | Model I | Model II | Model III | Model IV |
|---|---|---|---|---|
| zFDI_POP | 0.2757 * | | | 0.2059 |
| | (0.1267) | | | (0.1431) |
| zAMNW | −0.1983 ** | | | −0.0096 |
| | (0.0616) | | | (0.0938) |
| zIMMIG | −0.1836 | | | −0.3306 |
| | (0.1248) | | | (0.3405) |
| zNETMIG | −0.3769 * | | | −0.1941 |
| | (0.1641) | | | (0.2320) |
| zNETMIG_POP | 0.8424 *** | | | 0.6486 *** |
| | (0.1548) | | | (0.175) |
| zDBUS_POP | −0.039 | | | −0.0868 |
| | (0.0455) | | | (0.057) |
| zFRAG_ARABLE | | 0.2172 ** | | 0.0713 |
| | | (0.0665) | | (0.0522) |
| zFRAG_AGRI | | 0.0706 | | −0.0123 |
| | | (0.0724) | | (0.0535) |
| zAVER_SET | | 0.0419 | | 0.1377 |
| | | (0.2355) | | (0.2167) |
| zCENT_CITY | | −0.185 | | −0.0846 |
| | | (0.229) | | (0.2051) |
| CON (1–4) | | | −0.4024 ** | −0.3149 ** |
| | | | (0.1224) | (0.1323) |
| CON (1–9) | | | −0.5938 ** | −0.5659 ** |
| | | | (0.1730) | (0.1984) |
| EXEMP_VINEYARD | | | −0.0049 | 0.0025 |
| | | | (0.1631) | (0.1719) |
| Intercept | $-6.72 \times 10^{-17}$ | $-1.90 \times 10^{-16}$ | 0.3999 ** | 0.3515 ** |
| | (0.0439) | (0.0647) | (0.1161) | (0.1236) |
| Nº of observations | 410 | 410 | 410 | 410 |
| $R^2$ | | | | |
| within | 0.0830 | 0.0107 | 0.0000 | 0.0971 |
| between | 0.7397 | 0.3248 | 0.0000 | 0.7713 |
| overall | 0.2228 | 0.0709 | 0.0477 | 0.2445 |
| sigma_u | 0.0000 | 0.2895 | 0.3783 | 0.0000 |
| sigma_e | 0.8928 | 0.9208 | 0.9046 | 0.8841 |
| rho | 0.0000 | 0.09 | 0.1488 | 0.0000 |
| Wald $\chi^2$ | 115.52 *** | 16.74 ** | 23.85 *** | 128.17 *** |

Note: standard errors are given in brackets, * indicates significance level $\alpha < 0.05$, ** $\alpha < 0.02$, and *** $\alpha < 0.001$. Source: own calculation.

At first sight, a statistically significant inverse relationship between the average monthly nominal wage and the volume of withdrawn agricultural land is unclear. It can be seen as a reflection of the fact that income levels in Slovakia are higher in the urbanized regions, where the baseline area of the agricultural land per capita is lower. However, we cannot exclude the possibility that this relationship is more direct. The income level often reflects the overall level of financial capital in

a given territory. In this context, this may mean that entities in such regions have the opportunity to use alternative resources for areas with new features; because they have the resources that they can use e.g., for brownfield revitalization. This is considerably more expensive than building on the green field, but it reduces the pressure on the agricultural land [48].

Within the territorial characteristics, only the rate of arable land fragmentation was statistically significant, and with increasing parcel size, the withdrawal rate is increasing. We also explain this relationship through the financial costs, specifically transaction costs. In the case of any construction on the agricultural land, it is much cheaper for an investor, an owner or other relevant entity to obtain a unified land plot than to negotiate with several original owners. The Ministry of Agriculture and Rural Development of the Slovak Republic in 2019 identified an average of 12 co-owners per 1 land plot, while one owner owns 23 parcels [49].

In terms of legislative changes, the introduction of contributions on vineyards, or the introduction of exemptions for family houses does not statistically significantly affect the withdrawal rate. On the other hand, we also showed a positive impact of introducing the contributions on reducing the volume of the withdrawn agricultural land. For comparison, when withdrawal contributions were not collected, extending their collection to all quality groups (i.e., from 1 to 9) had a noticeably greater effect on reducing the volume of the agricultural land withdrawn than when they were collected only for the withdrawal of quality groups 1 to 4. As confirmed by Sobocká [41], the legislative change has increased the protection of the agricultural land from 21 to 37% for the highest quality agricultural land in each cadastral area.

When taking into account all factors (Model IV), only the positive impact of the legislative changes remains significant together with the migration balance per capita, which has a negative effect on the total volume of the land withdrawn.

In order to comprehensively understand the cause-and-effect relationships between the analyzed factors and the intensity of the withdrawal of agricultural land, we need to carry out more detailed analyses. Therefore, in the following sections, we estimate the relationship between all the factors analyzed and the volume of the land withdrawn for individual purposes.

## 3.1. The Volume of Agricultural Land Withdrawn for the Housing Purposes

As shown by the models' results (Table 2), only developmental and legislative factors are significant in the case of land withdrawal.

In terms of developmental factors, the highest withdrawal intensity has been demonstrated in the suburbanized areas and in the areas with the inflow of foreign direct investment. Contrarily, in the urbanized areas, there is lower land availability resulting in the lower occupation of the agricultural land. Urbanized areas offer alternative and more expensive housing options, which are, however, allowed by the higher income level of the population. Conversely, the lower-income population is to a greater extent migrating to urbanized areas and looking for housing opportunities in suburbanized areas, where the pressure on the land withdrawal is consequently higher [50].

This is also evidenced by a statistically significant positive coefficient of the share of migration balance per capita. Even in the case of land withdrawal for housing purposes, there has been a decreasing trend after the introduction of contributions, with a more substantial effect on this reduction being the extension of the obligation to pay contributions and for the withdrawal of agricultural land from lower quality groups, which was predicted by Sobocká [41].

Legislative factors have a significant impact on the withdrawal of land, but according to the opinion of experts, it is not a decisive factor in the choice of housing.

**Table 2.** Results of panel regression models where the dependent variable is the volume of agricultural land withdrawn for housing purposes.

| Factor: | Model I | Model II | Model III | Model IV |
|---|---|---|---|---|
| zFDI_POP | −0.0843 | | | −0.2356 |
| | (0.1264) | | | (0.1429) |
| zAMNW | −0.1171 | | | 0.0516 |
| | (0.0615) | | | (0.0936) |
| zIMMIG | 0.1061 | | | −0.0992 |
| | (0.1246) | | | (0.3399) |
| zNETMIG | −0.1046 | | | 0.1288 |
| | (0.1638) | | | (0.2316) |
| zNETMIG_POP | 0.5529 *** | | | 0.3932 * |
| | (0.1545) | | | (0.1747) |
| zDBUS_POP | 0.0222 | | | −0.0054 |
| | (0.0454) | | | (0.0569) |
| zFRAG_ARABLE | | 0.1519 | | −0.0503 |
| | | (0.0794) | | (0.0522) |
| zFRAG_AGRI | | 0.1019 | | 0.0534 |
| | | (0.0864) | | (0.0534) |
| zAVER_SET | | −0.0574 | | 0.1257 |
| | | (0.2811) | | (0.2164) |
| zCENT_CITY | | −0.0446 | | 0.0529 |
| | | (0.2733) | | (0.2048) |
| CON (1–4) | | | −0.4054 ** | −0.3500 ** |
| | | | (0.1199) | (0.1320) |
| CON (1–9) | | | −0.5181 ** | −0.4418 * |
| | | | (0.1696) | (0.1980) |
| EXEMP_VINEYARD | | | −0.0371 | −0.0671 |
| | | | (0.1599) | (0.1716) |
| Intercept | $-5.25 \times 10^{-18}$ | $-7.42 \times 10^{-17}$ | 0.3805 ** | 0.3369 ** |
| | (0.0438) | (0.0773) | (0.1185) | (0.1233) |
| Nº of observations | 410 | 410 | 410 | 410 |
| $R^2$ | | | | |
| within | 0.0227 | 0.0003 | 0.0000 | 0.0426 |
| between | 0.8193 | 0.1599 | 0.0000 | 0.8532 |
| overall | 0.2258 | 0.0408 | 0.0404 | 0.2471 |
| sigma_u | 0.0000 | 0.4024 | 0.4277 | 0.0000 |
| sigma_e | 0.8860 | 0.9064 | 0.8869 | 0.8723 |
| rho | 0.0000 | 0.1646 | 0.1887 | 0.0000 |
| Wald $\chi^2$ | 117.51 *** | 6.80 | 20.99 *** | 129.98 *** |

Note: standard errors are given in brackets, * indicates significance level $\alpha < 0.05$, ** $\alpha < 0.02$, and *** $\alpha < 0.001$. Source: own calculation.

### 3.2. The Volume of Agricultural Land Withdrawn for Industry Development Purposes

Land withdrawal for industrial purposes is significantly affected by all the examined factors. Regarding legislative changes, there is a comparable effect of introducing contributions on the withdrawal of agricultural land to the volume of the land withdrawn for industrial development, as was the case with the already analyzed models.

In terms of territorial factors, arable land fragmentation significantly affects the volume of the withdrawn land. As the average land size increases, the volume of the withdrawn agricultural land

also increases. This further confirms the assumption that investors, who prefer to occupy larger parcels of land, are searching for sites where they can acquire land with low transaction costs, whose amount is significantly affected by fragmentation and complex ownership relationships. At the same time, the experts underlined that higher land fragmentation, as a negative phenomenon, often hindering the usability of the land for large scale farming as well.

**Table 3.** Results of panel regression models where the dependent variable is the volume of agricultural land withdrawn for industry development purposes.

| Factor: | Model I | Model II | Model III | Model IV |
|---|---|---|---|---|
| zFDI_POP | 0.2893 | | | 0.3798 ** |
| | (0.1631) | | | (0.1641) |
| zAMNW | −0.2045 ** | | | −0.1446 |
| | (0.0706) | | | (0.1071) |
| zIMMIG | −0.1498 | | | −0.1663 |
| | (0.1661) | | | (0.391) |
| zNETMIG | −0.2039 | | | −0.1844 |
| | (0.2148) | | | (0.2659) |
| zNETMIG_POP | 0.4519 * | | | 0.3019 |
| | (0.2023) | | | (0.2012) |
| zDBUS_POP | −0.0044 | | | −0.0046 |
| | (0.0484) | | | (0.0606) |
| zFRAG_ARABLE | | 0.2785 *** | | 0.2552 *** |
| | | (0.0515) | | (0.0624) |
| zFRAG_AGRI | | 0.0368 | | −0.0212 |
| | | (0.0561) | | (0.064) |
| zAVER_SET | | −0.3499 | | −0.4297 |
| | | (0.1825) | | (0.2555) |
| zCENT_CITY | | 0.2669 | | 0.2957 |
| | | (0.1775) | | (0.2421) |
| CON (1–4) | | | −0.2998 ** | −0.2232 |
| | | | (0.1275) | (0.1419) |
| CON (1–9) | | | −0.5258 ** | −0.3993 |
| | | | (0.1804) | (0.2145) |
| EXEMP_VINEYARD | | | 0.0834 | 0.1831 |
| | | | (0.1700) | (0.1840) |
| Intercept | $-3.91 \times 10^{-17}$ | $-6.43 \times 10^{-17}$ | 0.3052 ** | 0.1941 |
| | (0.0629) | (0.0501) | (0.1142) | (0.1371) |
| Nº of observations | 410 | 410 | 410 | 410 |
| $R^2$ | | | | |
| within | 0.0452 | 0.0093 | 0.0000 | 0.0549 |
| between | 0.1677 | 0.4844 | 0.0000 | 0.4996 |
| overall | 0.0636 | 0.0854 | 0.0292 | 0.1318 |
| sigma_u | 0.2715 | 0.1078 | 0.3003 | 0.1573 |
| sigma_e | 0.9383 | 0.9537 | 0.9429 | 0.9366 |
| rho | 0.0773 | 0.0126 | 0.0921 | 0.0274 |
| Wald $\chi^2$ | 23.6 *** | 33.93 *** | 13.44 ** | 52.55 *** |

Note: standard errors are given in brackets, * indicates significance level $\alpha < 0.05$, ** $\alpha < 0.02$, and *** $\alpha < 0.001$. Source: own calculation.

In the case of the developmental factors, we observe a similar impact of immigration on the requirements for the use of space and on the withdrawal of agricultural land. However, we must

acknowledge that the relationship between these two processes is likely to work in the opposite direction (the inflow of investors will cause the withdrawal of agricultural land and at the same time it attracts labor force to the region). We also identify a statistically significant effect of higher income levels on the reduction of the volume of agricultural land withdrawn for the industrial development. We explain this by the fact that in the most developed regions of Western Slovakia, the local economy has already been "saturated" in the context of the development of industrial sectors. Those are mainly services that do not have the same space requirements, which are being developed. Additionally, industries with lower added value, typical for their location in less developed regions, are more demanding regarding the occupation of space compared to the light industry sectors, which are more concentrated in the more developed regions [51].

Foreign direct investments are a statistically significant factor after taking into account all analyzed factor groups (Model IV). In the same analysis, no legislative changes have been shown to be significant, indicating the failure of legislative measures in protecting the agricultural land when conflicts between the use of space for the agricultural production and the interest of investors arise.

When assessing the impact of the legislative factors, land withdrawal is differentiated by the industry size. While the contribution obligation is negatively affecting smaller investors, large strategic investments are exempted from paying the contributions. As several experts have stated, this approach of the state is not justifiable from an environmental point of view. The negative side is that the large investors do not have any additional compensatory obligation in relation to the withdrawn land.

## 3.3. The Volume of Agricultural Land Withdrawn for Transport Development Purposes

So far, the weakest interactions of the analyzed factors are recorded in case of the land withdrawal for the purpose of the transport development (Table 4).

The effects of neither territorial nor developmental factors on the volume of the withdrawn land for the transport development purposes are significant as a whole.

The only factor that has a statistically significant impact on the intensity of the land withdrawal for the purpose of transport is the introduction of contributions for its withdrawal, while even the extending of categories of land, for which these contributions are compulsory, does not affect this intensity as in previous cases. The explanation for these empirical findings is most likely that the development and construction of the transport network (especially the most important transport routes) is governed by completely different mechanisms than the rest of the withdrawal purposes. Planning of the development of the transport network and major routes is given by the Concept of Territorial Development of the Slovak Republic [52], which is designed and managed centrally. For these reasons, this process is very little influenced by the examined factors. Experts add that, in order to reduce the land use conflicts, cooperation between the agricultural land protection authorities, architects and authorizing authorities is essential.

**Table 4.** Results of panel regression models where the dependent variable is the volume of agricultural land withdrawn for transport development purposes.

| Factor: | Model I | Model II | Model III | Model IV |
|---|---|---|---|---|
| zFDI_POP | −0.1497 | | | −0.2387 |
| | (0.1404) | | | (0.1635) |
| zAMNW | −0.0028 | | | 0.1639 |
| | (0.0683) | | | (0.1069) |
| zIMMIG | 0.0684 | | | −0.0029 |
| | (0.1384) | | | (0.3893) |
| zNETMIG | 0.2081 | | | 0.3338 |
| | (0.1819) | | | (0.2650) |

**Table 4.** *Cont.*

| Factor: | Model I | Model II | Model III | Model IV |
|---|---|---|---|---|
| zNETMIG_POP | 0.0012 | | | −0.0907 |
| | (0.1716) | | | (0.2001) |
| zDBUS_POP | 0.0225 | | | 0.0035 |
| | (0.0505) | | | (0.0628) |
| zFRAG_ARABLE | | 0.0370 | | −0.0618 |
| | | (0.0610) | | (0.0608) |
| zFRAG_AGRI | | 0.0053 | | −0.0155 |
| | | (0.0664) | | (0.0623) |
| zAVER_SET | | −0.1995 | | −0.0705 |
| | | (0.2163) | | (0.2507) |
| zCENT_CITY | | 0.1649 | | 0.1132 |
| | | (0.2103) | | (0.2374) |
| CON (1–4) | | | −0.3841 ** | −0.4374 ** |
| | | | (0.1319) | (0.1463) |
| CON (1–9) | | | −0.3246 | −0.4157 |
| | | | (0.1865) | (0.2203) |
| EXEMP_VINEYARD | | | −0.0639 | −0.1669 |
| | | | (0.1758) | (0.1900) |
| Intercept | $2.74 \times 10^{-17}$ | $2.66 \times 10^{-17}$ | 0.3026 ** | 0.3913 ** |
| | (0.0486) | (0.0594) | (0.1114) | (0.1388) |
| Nº of observations | 410 | 410 | 410 | 410 |
| $R^2$ | | | | |
| within | 0.0120 | 0.0456 | 0.0000 | 0.0324 |
| between | 0.2886 | 0.0285 | 0.0000 | 0.3847 |
| overall | 0.0449 | 0.0043 | 0.0233 | 0.0768 |
| sigma_u | 0.0000 | 0.2155 | 0.1842 | 0.1068 |
| sigma_e | 0.9821 | 0.9626 | 0.9749 | 0.9620 |
| rho | 0.0000 | 0.0477 | 0.0344 | 0.0122 |
| Wald $\chi^2$ | 18.96 ** | 1.31 | 10.02 ** | 30.96 ** |

Note: standard errors are given in brackets, * indicates significance level $\alpha < 0.05$, ** $\alpha < 0.02$, and *** $\alpha < 0.001$. Source: own calculation.

### 3.4. The Volume of Agricultural Land Withdrawn for Mining Purposes

There are very similar interactions between the analyzed factors and the rate of withdrawal of agricultural land for the mining purposes as in the case of the total volume of the withdrawn land. This effect is even more visible in the case of the population movements measured by the migration balance. At the same time, we assume that the similar causal consequences that we considered at the beginning of this chapter also apply to the withdrawal for the mining purposes. Regarding the statistically significant inverse relation of the average monthly nominal wage and the volume of agricultural land for mining purposes, analogous to the previous interpretations, we assume that it is due to the nature of the localization behavior of the mining industries. These, of course, depend primarily on the spatial distribution of mineral resources, but they also tend to concentrate in less developed regions. These activities are space-demanding in the area they occupy. This was also confirmed by the finding that the larger average land size (namely arable land) leads to an increased land withdrawal frequency for mining purposes. In case of the withdrawal of land for the mining purposes, the relatively consistent impact can clearly be attributed to the foreign direct investments. What is positive, however, is that the introduction of contributions, as well as their extension to other land quality categories, has led to a reduction in the volume of the land withdrawal for the mining purposes. The experts confirmed that

the increase in the extraction of gravel and other minerals had a negative impact on the extent of the agricultural land withdrawn. Therefore, it is necessary that the mining industry, as well as the state, take responsibility for land use conflicts in the field of mining.

**Table 5.** Results of panel regression models where the dependent variable is the volume of agricultural land withdrawn for mining purposes.

| Factor: | Model I | Model II | Model III | Model IV |
|---|---|---|---|---|
| zFDI_POP | 0.4329 ** | | | 0.4730 ** |
| | (0.1561) | | | (0.1711) |
| zAMNW | −0.1942 ** | | | −0.0558 |
| | (0.068) | | | (0.1110) |
| zIMMIG | −0.230 | | | −0.2785 |
| | (0.1587) | | | (0.4080) |
| zNETMIG | −0.7248 *** | | | −0.6681 ** |
| | (0.2053) | | | (0.2767) |
| zNETMIG_POP | 1.032 *** | | | 0.8992 *** |
| | (0.1934) | | | (0.2106) |
| zDBUS_POP | −0.0288 | | | 0.0034 |
| | (0.0468) | | | (0.0584) |
| zFRAG_ARABLE | | 0.1825 ** | | 0.1069 |
| | | (0.0653) | | (0.0688) |
| zFRAG_AGRI | | 0.0497 | | −0.0409 |
| | | (0.0711) | | (0.0706) |
| zAVER_SET | | 0.1355 | | 0.0895 |
| | | (0.2313) | | (0.2766) |
| zCENT_CITY | | −0.2386 | | −0.1782 |
| | | (0.225) | | (0.2626) |
| CON (1–4) | | | −0.5727 *** | −0.5349 *** |
| | | | (0.1237) | (0.1378) |
| CON (1–9) | | | −0.6313 *** | −0.5460 ** |
| | | | (0.1749) | (0.2103) |
| EXEMP_VINEYARD | | | 0.0352 | 0.1068 |
| | | | (0.1649) | (0.1782) |
| Intercept | $6.64 \times 10^{-17}$ | $-1.91 \times 10^{-16}$ | 0.3423 ** | 0.4004 ** |
| | (0.0598) | (0.0636) | (0.1128) | (0.1388) |
| Nº of observations | 410 | 410 | 410 | 410 |
| $R^2$ | | | | |
| within | 0.0858 | 0.0087 | 0.0000 | 0.1120 |
| between | 0.3722 | 0.259 | 0.0000 | 0.4683 |
| overall | 0.1331 | 0.0501 | 0.0405 | 0.1786 |
| sigma_u | 0.2453 | 0.2758 | 0.2740 | 0.2347 |
| sigma_e | 0.8794 | 0.9413 | 0.9448 | 0.8729 |
| rho | 0.0722 | 0.079 | 0.0776 | 0.0674 |
| Wald $\chi^2$ | 50.95 *** | 12.26 ** | 18.58 *** | 71.77 *** |

Note: standard errors are given in brackets, * indicates significance level $\alpha < 0.05$, ** $\alpha < 0.02$, and *** $\alpha < 0.001$. Source: own calculation.

### 3.5. The Volume of Agricultural Land Withdrawn for Other Purposes

Non-agricultural purposes, for which agricultural land may be withdrawn from the agricultural land fund on request and which are not included in the categories of withdrawal we have already analyzed, are included in the group of other purposes. In order to comprehensively analyze the factors

that affect the volume of withdrawn agricultural land, but also to ensure that all factors are correctly identified, we also analyze this withdrawal category (Table 6). All models are significant as a whole, and a set of factors that have been identified as statistically significant is almost identical to a set of factors that determine the total volume of the withdrawn agricultural land. It should be noted, however, that in the case of land withdrawal for the non-categorized purposes, the positive effects of legislative changes (namely the introduction of contributions) on the intensity of withdrawal of agricultural land have only been manifested after their extension to the all land quality groups.

**Table 6.** Results of panel regression models where the dependent variable is the volume of agricultural land withdrawn for other purposes.

| Factor: | Model I | Model II | Model III | Model IV |
|---|---|---|---|---|
| zFDI_POP | 0.3990 ** (0.1489) | | | 0.3348 * (0.1649) |
| zAMNW | −0.2353 ** (0.0684) | | | −0.0549 (0.1076) |
| zIMMIG | −0.2361 (0.1492) | | | −0.7014 (0.3929) |
| zNETMIG | −0.4617 ** (0.1944) | | | −0.1370 (0.2672) |
| zNETMIG_POP | 0.7046 *** (0.1832) | | | 0.4604 * (0.2022) |
| zDBUS_POP | −0.003 (0.0486) | | | −0.0361 (0.0610) |
| zFRAG_ARABLE | | 0.1549 ** (0.0584) | | 0.1348 * (0.0627) |
| zFRAG_AGRI | | 0.0306 (0.0636) | | −0.0103 (0.0642) |
| zAVER_SET | | 0.2619 (0.2072) | | 0.4441 (0.2565) |
| zCENT_CITY | | −0.3574 (0.2015) | | −0.1257 (0.2431) |
| CON (1–4) | | | −0.3138 ** (0.1278) | −0.2780 (0.1427) |
| CON (1–9) | | | −0.4661 ** (0.1807) | −0.4365 * (0.2158) |
| EXEMP_VINEYARD | | | −0.101 (0.1704) | −0.0182 (0.1852) |
| Intercept | $6.04 \times 10^{-17}$ (0.0542) | $2.00 \times 10^{-17}$ (0.0569) | 0.3423 ** (0.1128) | 0.2913 * (0.1378) |
| Nº of observations | 410 | 410 | 410 | 410 |
| $R^2$ | | | | |
| within | 0.0541 | 0.0117 | 0.0000 | 0.0631 |
| between | 0.2933 | 0.2796 | 0.0000 | 0.4167 |
| overall | 0.0917 | 0.0441 | 0.0405 | 0.1198 |
| sigma_u | 0.1765 | 0.1984 | 0.2740 | 0.1570 |
| sigma_e | 0.9456 | 0.9599 | 0.9448 | 0.9463 |
| rho | 0.0337 | 0.041 | 0.0776 | 0.0268 |
| Wald $\chi^2$ | 36.5 *** | 13.45 ** | 18.58 *** | 48.53 *** |

Note: standard errors are given in brackets, * indicates significance level $\alpha < 0.05$, ** $\alpha < 0.02$, and *** $\alpha < 0.001$.
Source: own calculation.

Within V4 countries, similar comprehensive analysis was conducted in Poland by Sroka et al. [53] with somewhat comparable results. Although lacking the legislative aspects, with application of regression trees, they confirmed the high importance of population density and migration on differences in share of agricultural land in the overall surface of the municipal area, while the most important factor was the intensity of entrepreneurial activity, which we did not confirm. It should be noted, however, that they did not differentiate between foreign investments and local business activities.

## 4. Conclusions

The main problem of the current protection of the agricultural land in Slovakia is the conflict of interests in agricultural land use based on the diversity of needs of individuals on the one hand and the interest of the state and society on the other hand. The policy-making of the state is based on the assumption that the land protection can be ensured in particular by maintaining its area.

By examining the legislative development of land protection, the state mainly uses economic instruments to introduce and modify the withdrawal contribution obligation and regulates exemptions from it based on the state's current interests.

The research has shown that developmental factors for investment and development activities directly affect the total volume of the withdrawn agricultural land. From the aspect of the conflict of interests between individual and state regarding land protection, the private interest prevails over the public one. As a consequence, agricultural land is withdrawn in suburbanized and attractive areas where the land of the highest quality category is mostly located.

Aside from the industrial development, great conflicts arise between agricultural and residential use of the land. In many municipalities in Slovakia, there is still housing stock available, but nevertheless, most new residential areas are built on the agricultural land that has been withdrawn.

In accordance with the precautionary principle, the state should adopt a long-term conceptual document defining areas of agricultural land use taking into account the impact of developmental factors on the land protection. Similarly, the state should introduce new tools for land protection intended for entities not currently influenced by the introduced legislative instruments.

**Author Contributions:** Conceptualization, I.M., L.P., K.M.; methodology, K.M.; validation, I.M., L.P., K.M.; formal analysis, I.M.; investigation, I.M., L.P., K.M.; resources, I.M.; data curation, I.M, K.M.; writing—original draft preparation, I.M., L.P., K.M.; writing—review and editing, I.M., L.P., K.M.; visualization, I.M.; supervision, I.M., L.P., K.M.; project administration, L.P.; funding acquisition, L.P.

**Funding:** This research was funded by the Scientific Grant Agency project "Protection of conservation of agricultural land acreage in Slovakia", No. V-18-015-00, with the support of the Ministry of Education, Science, Research and Sport of the Slovak Republic and the Slovak Academy of Sciences.

**Conflicts of Interest:** The authors declare no conflict of interest. The sponsors had no role in the design, execution, interpretation, or writing of the study.

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
