# Peer review of "Modelling Development, Territorial and Legislative Factors Impacting the Changes in Use of Agricultural Land in Slovakia"

_sustainability, doi:10.3390/su11143893_

Round 1

Reviewer 1 Report

Conflicts in Agricultural Land Use – Modelling Factors Impacting the Changes in Use of Agricultural Land

GENERAL COMMENTS AND SUGGESTIONS:

1.   This article aims to identify factors affecting the withdrawal of agricultural land and to initiate a professional discussion on the concept of the protection and use of agricultural land  in Slovakia.

2.   As part of the research, the development of territorial and legal factors influencing the withdrawal of land for residential, industrial, transport, mining and other purposes was analyzed, using panel regression models.

3.   Lack of references in the article to other EU countries in the aspect of excluding land from production. The review of the literature should include items with analogical themes, especially those published in the journal Sustainability. Please look at the number (Sustainability, 2018 vol. 10 iss. 1) and comment on the analogies or differences with at least one of the neighboring countries to Slovakia (at least Poland and the Czech Republic).

https://doi.org/10.3390/su10010136

4.   In Conclusion, the development of the Slovakia should be summarized in the aspect of land use change in the context of neighboring countries (Poland, Czech Republic), in which political system transformation also took place.

Author Response

Dear Reviever,

thank you very much for your comments and suggestions.

We tried to do our best to fulfill your expectations.

Kindest regards,

Luca Palšová,

Katarína Melichová,

Ina Melišková

Reviewer 2 Report

The problem is that citations are biased towards textbooks and administrative documents and there are few references to scientific knowledge. I read that journal of sustainability is read by researchers in a wide range of research fields, so I think it needs clear explanations. We can underatand the problem of the current protection of agricultural land in Slovakia is the conflict of interests in farm land use. However, we can not think that this manuscript has little scientific knowledge.

Detail:

41L:soils of average quality prevail? It is difficult to understand.
51L: loss of soil source? Sorry, Please explain the scientific basis more carefully. I think it is necessary to refer academic papers.
53L: Are there other similar constitutional countries? It is important to show how this research can be used in many countries.
106L: Where is data source (country statical data?). Please refer it.
141L: Is this original method? Please introduce the appropriate previous study.
182L: As above.

455-500:These sentence may be hard to read because there are many line breaks.

Author Response

(The authors gave the same response as above.)

Reviewer 3 Report

General comments

This manuscript by Palšová et al. studied the factors affecting agricultural land loss and the conflict between urban development and agricultural land use in Slovakia from 2007 to 2016. The rationale behind this research is clear and the importance of such studies is crucial. However, the methods used are pure statistics. Why the authors did not consider using remotely sensed data or at least available high-resolution land cover maps for Europe? The results need to be presented in a more effective and interesting fashion. Because, as it stands, it is presented in a not interesting manner with several large tables. Furthermore, minor English language revision is required and there are several major/minor corrections and clarifications required to be addressed to improve the quality of the manuscript. Currently, this manuscript is not sufficiently ready to be published at Sustainability. Please see my detailed comments below.

Specific Comments

Title: The title is generic and I feel that it does not reflect the methods used in the manuscript.

Abstract: Line 16: What do you mean by (withdrawal of agricultural land)? Do you mean abandonment/decline of agricultural land or loss due to urban development? Moreover, the authors did not mention any concluded quantitative results related to the amount of agricultural land lost.

Introduction: Line 31: EU should be defined the first time it is mentioned as (European Union).

Introduction: Lines 50-52: It is necessary to expand a little more the literature review demonstrating the importance of such studies worldwide. The citations included in this paper are local and may lead the readers to believe that this is only a local perspective. There are several studies addressing these issues worldwide. These are some recent references examples in Asia and Africa, which might be useful to look at and include (Shi et al. 2016 in sustainability; Rimal et al. 2018 in land; Rimal et al. 2018 in sustainability; Radwan et al. 2019 in remote sensing).

Introduction: Too many details about the local acts and laws, which are possibly unknown for anyone from elsewhere outside Slovakia. Furthermore, the authors are advised to add a final paragraph in the introduction mentioning the objectives of the manuscript in a clear and straightforward fashion.

Materials and Methods: Lines 107-112: These lines should be moved to the introduction as the last paragraph demonstrating the aim of the paper.

Materials and Methods: Line 111: Why did you choose this specific time period 2007-2016?

Materials and Methods: Line 126: What do you mean by (Observations) here?

Results and Discussion: Line 194: Figure 1: Units must be added to the axes of the chart.

Results and Discussion: Line 190: The presentation fashion of the results is boring I am afraid. With six complicated large tables, which make the reader not interested to follow the story behind the paper. However, figure 1 and figure 2 were good and interesting and I was expecting the rest of the data to be presented in a similar fashion. Furthermore, I suggest writing the discussion section separately.

Conclusions: Line 440: A lot of repetitions from the abstract, introduction and methods. This section is too lengthy (66 lines!) and needs to be re-written in a more concise fashion. I would say 15-20 would be enough. Furthermore, the authors did not mention any concluded quantitative results related to the amount of agricultural land lost.

References: Line 517: As mentioned before, the authors presented their research in a local way. Most of the references cited are relevant to Slovakian authorities/researchers. This needs to be expanded to include further similar studies in Europe and globally as well. Furthermore, I am wondering if citing 27 references is sufficient for such a nation-based study.

Author Response

(The authors gave the same response as above.)

Round 2

Reviewer 2 Report

I think this manuscript can accept in present form.

Author Response

Dear Reviewer,

thank you very much for your effort to review our article.

We appracite all your comments that improved the quality of the article.

Our kindest regards,

Lucia Palšová,

Ina Melišková,

Katarína Melichová

Reviewer 3 Report

I would like to thank the authors for their efforts in improving the manuscript. The current iteration is more improved and the text has sufficient details in most sections. However, I still believe that the authors need to address the following before considering the manuscript for publication at sustainability:

1 - I suggest the title to be something like (Modelling Development, Territorial and Legislative Factors Impacting the Changes in Use of Agricultural Land in Slovakia). This is more straightforward now. otherwise, you can consider removing Slovakia and putting it in the keywords along with (European Union) and you might also consider removing the last 3 keywords because they already exist in the current title.

2 - I feel that the introduction section is a bit lengthy.

3 - In my previous review, I suggested the authors separate the discussion section but it seems that they didn't. I think separating the discussion section will make the context of the paper clearer and remove any complexity from the (not very interesting) results section.

4 - I still believe that the conclusions section is lengthy (38 lines). I would say 20 - 25 lines max should be sufficient.

Author Response

Dear Reviewer,

thank you very much for your effort to review our article.

We appreciate all your comments that improved the quality of the article.

Our kindest regards,

Lucia Palšová,

Ina Melišková,

Katarína Melichová

Round 3

Reviewer 3 Report

I would like to thank the authors for their efforts in improving the quality of their manuscript. The current iteration is more improved and the text has sufficient details in most sections. The manuscript now is ready to be considered for publication at sustainability.

Just a final suggestion to improve the readability of the title. It could be (Modelling Development, Territorial and Legislative Factors Impacting the Agricultural Land Use Changes in Slovakia).